# Prognostic Value of TNFR2 and STAT3 among High-Grade Serous Ovarian Cancer Survivors According to Platinum Sensitivity

**DOI:** 10.3390/diagnostics11030526

**Published:** 2021-03-16

**Authors:** Janisha Silva Raju, Nor Haslinda Abd. Aziz, Ghofraan Abdulsalam Atallah, Chew Kah Teik, Mohamad Nasir Shafiee, Muhammad Fakhri Mohd Saleh, Ravichandran Jeganathan, Reena Rahayu Md Zin, Nirmala Chandralega Kampan

**Affiliations:** 1Department of Obstetrics and Gynaecology, Universiti Kebangsaan Malaysia Medical Centre, Kuala Lumpur 56000, Malaysia; janisha82@gmail.com (J.S.R.); norhaslinda.abdaziz@ppukm.ukm.edu.my (N.H.A.A.); ghofraan.a@gmail.com (G.A.A.); drchewkt@gmail.com (C.K.T.); nasirshafiee@hotmail.com (M.N.S.); 2Department of Pathology, Universiti Kebangsaan Malaysia Medical Centre, Kuala Lumpur 56000, Malaysia; mfms0209@gmail.com (M.F.M.S.); reenarahayu@ppukm.ukm.edu.my (R.R.M.Z.); 3Department of Obstetrics and Gynaecology, Hospital Sultanah Aminah Johor Bahru, Johor Bahru 80000, Malaysia; drjravi@gmail.com

**Keywords:** tumour necrosis factor receptor 2 (TNFR2), signal transducer and activator of transcription 3 (STAT3), high-grade serous ovarian cancer (HGSC), platinum sensitive (PS), platinum resistant (PR), progression free survival (PFS)

## Abstract

This study’s goal was to determine the protein expression level of tumour necrosis factor receptor 2 (TNFR2) and signal transducer and activator of transcription 3 (STAT3) in high-grade serous ovarian cancer (HGSC) tissues in relation to the platinum-based chemotherapy response and the prognosis outcome. A total of 25 HGSC patients underwent primary surgical debulking followed by first-line adjuvant platinum-based chemotherapy. Tissue microarray (TMA) slides were constructed utilising archived formalin fixed paraffin embedded (FFPE). The protein expression of TNFR2 and STAT3 were analysed using immunohistochemistry (IHC) staining and subsequently were correlated to the clinicopathological characteristics, platinum sensitivity as well as the duration of progression-free survival. About 14 out of 25 patients (56.0%) were platinum-sensitive. The progression free survival was significantly longer in the platinum-sensitive (PS) group when compared to those with the platinum-resistant group (PR), *p* = 0.0001. Among patients with TNFR2 strong expression on ovarian tissue, there was a significantly longer progression-free survival interval of 540 days in the PS group compared to PR, *p* = 0.0001. Patients with STAT3 expression also showed significantly better progression-free survival of 660 days in the PS group when compared to the PR group, *p* = 0.0001. In conclusion, patients with strong TNFR2 and STAT3 expression in the ovarian tissue had significantly longer progression-free survival interval in the PS group. Nevertheless, further research with a larger number of tissues may be required to demonstrate further significant differences.

## 1. Introduction

The American Cancer Society predicts that an estimated 21,750 new cases of ovarian cancer will be reported in 2020 and 13,940 people will suffer from ovarian cancer in the United States with just 48.6% having a 5-year survival rate [1]. As there are no specific clinical symptoms, epithelial ovarian cancer is generally diagnosed at an advanced stage (stage III and stage IV) in up to 75% of patients, and the commonest type is high-grade serous ovarian cancer (HGSC) [2]. Unfortunately, the effectiveness of screening to detect the disease at an earlier and curable stage remains unproven.

The standard of care for patients with HGSC consists of maximal cytoreductive surgery and platinum-based chemotherapy treatment [3]. Most of the patients (75%) reported HGSC at diagnosis experience a good initial response to optimal cytoreductive surgery followed by adjuvant platinum-based chemotherapy [4]. Unfortunately, up to 80% of HGSC patients establish tolerance to traditional chemotherapy leading to disease recurrence, hence posing a great challenge to subsequent chemotherapy choices [4,5]. Furthermore, the overall 10-year survival in HGSC following standard therapy is poor at only 30% [5].

In disease recurrence, the time interval from last dose of treatment with platinum-based compounds is prognostic and correlated to improved survival. Relapsed disease is usually managed with a chemotherapy regimen; the choices are based on duration elapsed after the previous dose of a first-line platinum-based compound therapy. Patients that have relapsed more than 6 months after their last dosage of platinum-based therapy are categorised as platinum-sensitive and have a 60% chance of response to platinum therapy [6,7]. On the other hand, patients that have relapsed with a platinum-free period of fewer than 6 months are graded as platinum-insensitive and are not rechallenged with platinum-based regimens due to low treatment response [3]. Hence, the platinum-resistant patients would benefit from alternative therapy or targeted therapy, such as immunotherapy.

Antibody immunotherapy is showing a great potential in the management of cancer. It has been found that strong expression of the tumour necrosis factor receptor 2 (TNFR2) on regulatory T cells (Tregs) in ovarian cancer tissue creates a potent immunosuppressive tumour microenvironment and is associated with poor clinical response [8,9]. The use of antagonistic antibodies to TNFR2 that will target the TNFR2-positive cancer cells may help to inactivate TNFR2-positive Tregs (TNFR2^+^ Tregs) as observed by Torrey et al. [10]. C. Govindaraj et al., on the other hand, discovered that targeting and reducing the highly immunosuppressive TNFR2^+^ Tregs within ovarian cancer patients, could potentially disrupt numerous immunoregulatory circuits within the tumour microenvironment and therefore improve response to treatment [11].

The Janus Kinase/Signal transducer and activator of transcription 3 (JAK/STAT3) pathway is one a major signalling pathway that is critical for ovarian tumour growth [12]. Signal transducer and activator of transcription 3 (STAT3) is predominantly located in an inactive state within the cytoplasm and is activated following phosphorylation at Tyr705 by the Janus family kinases [13]. The phosphorylated STAT3 (*p*-STAT3) protein may then be transported to the nucleus where it attaches to DNA and stimulates the transcription of numerous genes that control critical cell activity, including cell survival, proliferation, new vessel formation and tumour evasion. The activation of this pathway is found to be associated with tumour progression and resistance to chemotherapy, as well as poor prognosis in patients with ovarian cancer [12]. Rosen et al. detected that the activation and translocation of *p-STAT3* to the nucleus were associated with a poor prognosis in ovarian cancer [14]. Yang et al. reported that increased *p-STAT3* expression in the omental tissue correlated with poor survival amongst patients with HGSC [15].

In this study, we analysed the protein expression level of TNFR2 and STAT3 among our chemo-naïve patients with HGSC, according to their response to platinum-based chemotherapy, either platinum-sensitive or platinum-resistant. The findings of this study could enable early prediction of the response to first-line platinum-based chemotherapy, and segregate those chemo-resistant patients who might benefit from an alternative individualised therapy.

## 2. Materials and Methods

This is a retrospective cohort study concerning patients with high-grade serous ovarian carcinoma (HGSC), treated at the Department of Obstetrics and Gynaecology, Universiti Kebangsaan Malaysia Medical Centre (UKMMC) from August 2011 to August 2019. In addition, this study utilised paraffin block samples of HGSC tissues archives from the Department of Pathology UKMMC that were collected from August 2011–October 2018. This study was approved by the Universiti Kebangsaan Malaysia medical ethics committee, with the project code FF-2019-278.

Our aim is to determine the protein expression level of TNFR2 and STAT3 in tumour tissues collected from chemo-naïve patients diagnosed with high-grade serous ovarian carcinoma (HGSC) and to evaluate its association with platinum-based chemotherapy response and the prognosis outcome via progression-free survival. The response to platinum chemotherapy is considered platinum-sensitive (PS) in those patients without recurrence or disease progression at 6 months and above from the last course of platinum chemotherapy. In contrast, patients with disease recurrence within 6 months after treatment are considered the platinum-resistant (PR) group.

The study subjects were women with HGSC who underwent primary cytoreductive surgery for ovarian tumours, followed by adjuvant platinum-based chemotherapy. The treatment consists of intravenous carboplatin AUC 5–6 and intravenous paclitaxel 175 mg/m^2^ every 21 days for 6 cycles. None of the patients had maintenance therapy such as bevacizumab or PARP inhibitor, although offered, due to the cost as that was self-funded therapy. All patients underwent 6 cycles of chemotherapy with no significant adverse reaction. We excluded patients with HGSC receiving neo-adjuvant chemotherapy prior to undergoing interval debulking surgery. The ovarian tissues belonging to patients with severe pre-existing medical conditions (such as cardiac, liver or vascular diseases) and those with open biopsy, major operation, trauma or disability within 28 days prior to surgery as well as use of NSAIDS, anti-inflammatory steroids and/or immunosuppressive agents within 14 days prior to surgery were also excluded from further analysis. In addition, those with missing or incomplete medical records and those with defective paraffin blocks or having no extra paraffin blocks in reserve were eliminated from analysis.

Patients were followed up every 3–4 months in the Ovarian Cancer Clinic and all patients had at least a completed year of follow-up. During each visit, the patients were enquired about clinical symptoms and examined to detect recurrence as well as blood sampling for serials CA-125 levels. Radiological imaging was performed to confirm the presence of recurrence if any sign was visible with the clinical or biochemical examination. Those with disease progression during chemotherapy were also included in the resistant group. Progression-free survival (PFS) was defined by the time from the initial treatment intervention (surgery or chemotherapy) to evidence of disease progression. Progressive disease (PD) was defined by either confirmed increased of serum CA125 levels to more than twice the nadir or upper limit of normal (>35 IU/mL), documentation of RECIST-based progression of old lesions or appearance of new lesions on radiological imaging, or death.

### 2.1. Tissue Microarray (TMA) Construction Technique

Formalin-fixed, paraffin-embedded tissue blocks were retrieved along with their corresponding hematoxylin and eosin (H&E) stained slides. The desired tissue blocks (donor block) were chosen to construct tissue microarray (TMA) blocks using a tissue microarrayer, Alphelys Minicore 3 Tissue Arrayer (Alphelys, Plaisir, France). Different normal tissues were used to allow the identification of every row based on the morphology and to facilitate a simple orientation of rows. In addition, a set of orientation cores were placed at the top left of the array to orient the whole specimen.

Each tissue (donor) block was extracted from a single representative of 1.0 mm in diameter of the tissue heart. These tissue cores were arranged in hollow paraffin blocks (recipient block) of 28 × 22 mm, with a distance of 2.0 mm between the cores; making a limit of 8 × 8 dots in the recipient block. The TMA blocks were then heated to 60 °C for 5 min to mix the donor cores with the recipient stone. Finally, parts of the TMA blocks were sliced to 3μm and placed on adhesive coated slides for staining purposes.

### 2.2. Immunohistochemistry (IHC) Staining Method

Immunohistochemistry staining was conducted on tissue microarray parts using the EnVisionTM FLEX Mini Pack, High pH procedure (Code No. K8023, Dako, Glostrup, Denmark). Primary antibody was diluted to an acceptable concentration utilising Antibody Diluent, Dako REALTM (Code No. S2022, Dako, Glostrup, Denmark). Washing measures between each reagent were conducted using EnVisionTM FLEX Wash Buffer 20X (Code No. K8007, Dako, Glostrup, Denmark) diluted with 1X deionized water solution. 1X DAB-containing (Diaminobenzidine) Substrate Working Solution was prepared by diluting the 50× EnVisionTM FLEX DAB+ Chromogenic concentrate with EnvisionTM FLEX TM Substrate Buffer (Code No. K8023, Dako, Glostrup, Denmark).

Tissue microarray blocks were sliced roughly 3 μm thick and placed on the Platinum Pro White adhesive glass slide (Product No: PRO-01, Matsunami, Kishiwada, Japan). The slides were then air-dried at room temperature overnight followed by incubation at 60 °C on a hot plate for 1 h. An initial deparaffinization and antigen retrieval stage was performed in Decloaking ChamberTM NxGen (Ref. No: DC2012-220V, Biocare Medical, Pacheco, CA, USA) with temperature conditions of 110 °C within 30 min accompanied by cooling for 30 min at room temperature and washing with running tap water for 3 min. Subsequently, the slides were incubated with EnVisionTM FLEX Peroxidase-Blocking Reagent (Code No. DM821, Dako, Glostrup, Denmark) for 10 min, accompanied by a rinsing phase.

Slides were then incubated with primary antibodies (Table 1) at room temperature, accompanied by incubation with EnVisionTM FLEX/HRP (horseradish peroxidase) (DM822, Dako, Glostrup, Denmark) for 30 min. Parts were eventually incubated with 1X DAB-containing Substrate Working Solution for a duration of 10 min. Hematoxylin 2 (REF 7231, ThermoScientific, Waltham, MA, USA) was then dehydrated for 5 s, accompanied by a dehydration stage with increased alcohol solutions (80%, 90%, and 100% and 2-fold xylene. The slides were eventually assembled using the CoverSealTM-X xylene-based mounting medium (Cat. No.: FX2176, Cancer Diagnostics, Cambridge, MA, USA).

### 2.3. TNFR2 and STAT3 Protein Expression Scoring on TMA Blocks

The TMA cores were analysed by two researchers independently. All samples were scored for the expression of TNFR2 and STAT3. Cytoplasmic staining of TNFR2 is considered as positive; nuclear staining of STAT3 is considered as positive.

The interpretation of staining was blinded from the clinical outcome data before analysis. Scoring was interpreted in accordance to the fraction (in percentage) of positively stained tumour cells and then classified into the TNFR2-positive cells and STAT3-positive cells. The proportion score represents the estimated fraction of TNFR2-positive staining tumour cells and STAT3-positive staining tumour cells. The group was categorised as strong positive if more than 50% of tumour cells were expressed, whereas the weak positive was recorded when 5–50% of tumour cells were positively stained, and cells with less than 5% of the stain expression were grouped as negative (Figure 1 and Figure 2).

The TNFR2 and STAT3 protein expression levels among these patients with HGSC were compared against their clinic-pathological characteristics and platinum sensitivity. Their correlation with progression-free survival (PFS) was analysed individually.

### 2.4. Statistical Analysis

Statistical investigation was conducted using the Social Sciences Statistical Package for Windows (SPSS 24.0). Median, interquartile range (IQR) and frequencies (percentage) were used to describe the characteristics and biography of the participants. Specimens were categorised into two groups—PS and PR. Any demographic and clinical findings differences were analysed between the two treatment groups. These groups were then analysed against the TNFR2 and STAT3 expressing tissue level and correlated with progression-free survival (PFS). The median values of continuous variables within normal distribution between the two groups were compared accordingly using Independent Sample *t*-test. To analyse the inter-relationship of categorical data the chi-square test and Fisher’s exact probability test were utilised. In addition, the logistic regression analysis was used for data of more than three groups. The Kaplan–Meier survival curve analysis was used to calculate the progression-free survival interval between the PS and PR groups, as well as to highlight the relationship between the strength of TNFR2 or STAT3 expression groups in correlation to their platinum response. The statistical significance threshold is set at *p* value of 0.05 or lesser.

## 3. Results

A total of 32 patients were initially found eligible; however, a further six samples were excluded as the patients received neoadjuvant chemotherapy prior to their interval debulking surgery, and one was excluded as the paraffin block was defective. Hence, a total of 25 specimens were used in this study. Each of the 25 specimens was further analysed using three different tissue microarray (TMA) cores, making a total of 75 tissue microarray cores. A total of 14 out of 25 (56.0%) patients were platinum-sensitive (PS), while 11 patients (44.0%) were platinum-resistant (PR).

The demographic distribution is shown in Table 2. The mean age of diagnosis was 50.1 ± 13.4 years in the PS group, while the mean age was higher in the PR group (62.0 ± 6.4 years). The majority of PR patients were in advanced stage compared to those in the PS group. In addition, the PR occurred more commonly among the older age group 56–79 years old; *p* = 0.008. The mean age of menarche was at 12.9 ± 0.9 years among those with PS and 13.1 ± 0.9 years with PR. A majority of patients were in the BMI overweight category as shown in Table 2. There was no difference in the mean BMI of women with PS (26.0 ± 5.5 kg/m^2^) compared to those with PR (26.2 ± 5.8 kg/m^2^), *p* = 0.99.

The number of menopaused women in the PS group was 7 out of 14 and the rest were pre-menopausal (50%), while in the PR group, all patients (100%) were menopaused. Three patients among those in the PS group had a family history of breast, ovarian or colon cancer, whereas in the platinum-resistant group, only one patient had such a family background. One patient of the entire cohort had a personal history of breast cancer and was platinum-resistant.

As for the different FIGO (International Federation of Gynaecology and Obstetrics) stages of ovarian cancer, six out of 14 patients (42.9%) were at the early stage of the disease (Stage IC and II) in the PS group, while only one patient had Stage IC among those in the PR group. On the other hand, the majority of patients were at the advanced stage of the disease (stage III and IV), which accounts for 50% and 45.5% in the PS and PR group, respectively. There was no significant difference in the mean preoperative CA125 level between the PS (1389 ± 1575, range: 21–5631 IU/mL) and PR group (1566 ± 2107, range: 48–5857 IU/mL), *p* = 0.64.

The progression-free survival interval between the PS and PR group was demonstrated in Figure 3. The median progression-free survival interval in the PS group was significantly longer at 18 months compared to three months in those with PR, *p* = 0.0001.

The TNFR2 expression in the PS and PR groups is shown in Table 3. Overall TNFR2 protein has shown strong positive expression in 19 patients and weak positive expression in the rest of the six patients. Strong expression of TNFR2 was seen in both the PS (71.4%) and PR groups (81.8%). The difference of TNFR2 expression between the PS and PR groups was statistically not significant. On the other hand, the STAT3 protein showed strong positive expression in 19 patients and weak positive expression in 6 patients. Strong expression of STAT3 was seen in both the PS (78.6%) and PR groups (72.7%). Similarly, the difference of STAT3 expression between the PS and PR groups was statistically not significant (Table 4).

The protein expressions were analysed in correlation to the FIGO stages of ovarian cancer (Table 5). TNFR2 and STAT3 showed stronger expression in advanced stage disease (stage III and stage IV). In specific, both of the proteins showed stronger positive expression in stage III ovarian cancer.

The protein expression and its association with progression-free survival (PFS) is shown in Table 6. The PFS was longer in the weaker protein expression of both TNFR2 and STAT3 (TNFR2: 31 vs. 18 months, STAT3: 34 vs. 18 months); however, no significant difference was observed between the level of expression of the protein markers and the progression-free survival.

The progression-free survival interval among the TNFR2 positive patients in association to platinum sensitivity or resistance is demonstrated in Figure 4. Patients with TNFR2 expression had a longer median PFS interval of 540 days in the PS group, and a shorter interval of 90 days in the PR group, *p* = 0.0001. On the other hand, patients with STAT3 expression also showed a better PFS median of 660 days in the PS group, and a shorter interval of 120 days with PR, *p* = 0.0001 as shown in Figure 5. 

## 4. Discussion

Ovarian cancer is thought to be the most fatal cause of death for gynaecological malignancies, usually due to advanced stage at clinical presentation. Following cytoreductive surgery and conventional chemotherapy, most patients achieve initial good treatment response, but then succumb to recurrence and subsequent chemo resistance, so the overall survival rate remains poor. Given the heterogeneity and immunogenicity of HGSC, improvement in existing survival rates may be achieved by identifying immune biomarkers to aid treatment response and to further personalize individual strategies for treatment. New regimens targeting the pathway of metastases and chemo resistance are important for the production of more successful therapies for ovarian cancer, especially in the platinum chemo-resistant individuals.

The demographic differences between the PS and PR groups were not statistically significant except for the older age at diagnosis, the menopausal state and the poorer progression-free survival interval for those with PR. Hence, no difference suggests association of platinum sensitivity or resistance with the stage of HGSC of the ovary, the BMI of the patient or even the CA125 level. There were also no significant differences in the surgical debulking status between the two study groups with a majority having optimal surgery. Although not significant, it was noted that a majority of patients in the PR group were in advanced stages (Stage III-IV) in comparison to the PS group and this may contribute to poorer progression-free survival. A larger sample or a more homogenous study population including only advanced patients may help to show further clarity in future studies.

Although previous studies demonstrated the correlation between strong expression of TNFR2 with chemo resistance and poor progression-free survival [11,15], in our present study the presence of strong expression of TNFR2 was found to be high in both the PS and PR groups (81.8% vs. 71.4%, respectively), *p* = 0.661. Nevertheless, we were able to deduce a correlation between the strong expression of TNFR2 in the advanced stage of the disease. A total of 12 patients (48%) showed strong expression of TNFR2 with FIGO stage III and stage IV cancer. Strong expression of TNFR2 can be correlated to advanced stages of the disease, which carries a poorer clinical outcome; that also may contribute to the disease progression.

The activation of the JAK/STAT3 pathway, reported to play an important role in multiple oncogenic processes, including tumour proliferation, differentiation, angiogenesis and survival, has a significant impact on the survival of ovarian cancer patients [13,14]. Previous studies have shown total STAT3 and phosphorylated STAT3 (*p-STAT3*) are overexpressed in a subset of chemoresistant ovarian cancer cell lines as compared to their expression in the corresponding chemo-sensitive cell lines [13,14]. In comparison to the previous research, our present study demonstrated no significant difference in the presence of strong STAT3 expression between the PS and PR groups (78.6% vs. 72.7%, respectively, *p* = 0.514). Similar to strong TNFR2 expression, strong STAT3 expression was found in 15 patients (60%) with FIGO stage III and stage IV cancer. Additionally, about 50% of weak positive expression was found in FIGO stage 1C cancer. This finding may propose a good clinical response with STAT3 inhibition targeted treatment in early stage cancer, which carries a better prognostic outcome and survival. In addition, studies have also shown that STAT3 in conjunction with paclitaxel therapy synergistically reduced peritoneal seed and extended survival in the murine model of intraperitoneal ovarian cancer [16].

Progression-free survival and overall survival are important for evaluating the full impact of any new treatment. The PFS was significantly better in the PS compared to the PR group (median of 18 months vs. 3 months, *p* = 0.0001). The PFS interval was better in the weakly expressed compared to the strong expression group among both the TNFR2 (31 vs. 18 months, *p* = 0.743) and the STAT3 (34 vs. 18 months, *p* = 0.693) proteins markers but showed no significant difference. In TNFR2 strong expression, there was a significantly longer PFS interval of 540 days in the PS group, compared to only 90 days in the platinum-resistant group (*p* = 0.0001). Similarly, the STAT3 strong expression showed significantly better PFS of 660 days in the PS group, compared to 120 days in the platinum-resistant group with (*p* = 0.0001). The PFS interval in the PS group, was a better trend in the TNFR2 weak expression group with an interval of 585 days, compared to the TNFR2 strong expression group with an interval of 540 days (*p* = 0.58). However, this result was not statistically significant due to the small sample size. The limitation to this analysis is that the number of patients with weak TNFR2 or STAT3 signalling was small, which may be subject to bias and occurrence by chance. A larger sample size is required to further validate these findings. Nevertheless, the PFS interval was better among PS with TNFR2 or STAT3 weak expression tumours. Hence, PFS may be improved by targeting these protein markers in ovarian cancer treatment.

## 5. Conclusions

High-grade serous ovarian cancer tissue of chemo-naïve patients demonstrated strong expression of TNFR2 and STAT3 protein. The progression-free survival was significantly better in the platinum-sensitive in comparison to the platinum-resistant group. In both TNFR2 and STAT3 strong expression groups, there was a significantly longer progression-free survival interval in the PS compared to the PR group. Nevertheless, further research with a larger number of tissues may be required to demonstrate further significant differences.

## Figures and Tables

**Figure 1 diagnostics-11-00526-f001:**
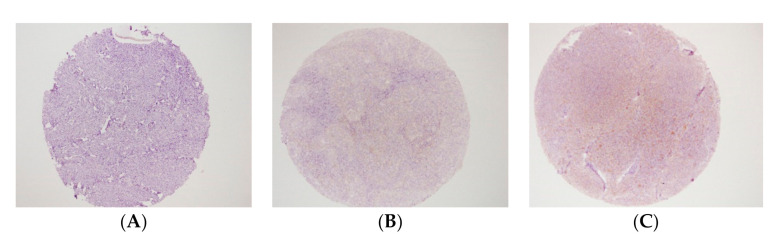
Immunohistochemistry staining of tumour necrosis factor receptor 2 (TNFR2). Under magnification of ×10. (**A**) Negative TNFR2 expression; (**B**) Weak positive TNFR2 expression; (**C**) Strong positive TNFR2 expression (intense cytoplasmic staining).

**Figure 2 diagnostics-11-00526-f002:**
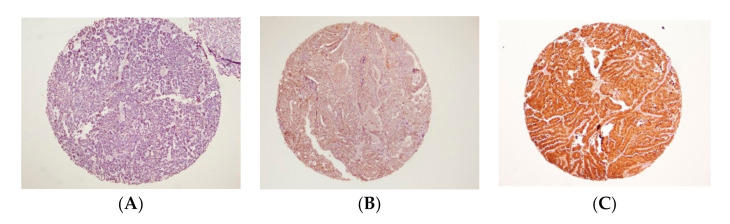
Immunohistochemistry staining of signal transducer and activator of transcription 3 (STAT3). Under magnification of ×10. (**A**) Negative STAT3 expression; (**B**) Weak positive STAT3 expression; (**C**) Strong positive STAT3 expression (intense nuclei staining).

**Figure 3 diagnostics-11-00526-f003:**
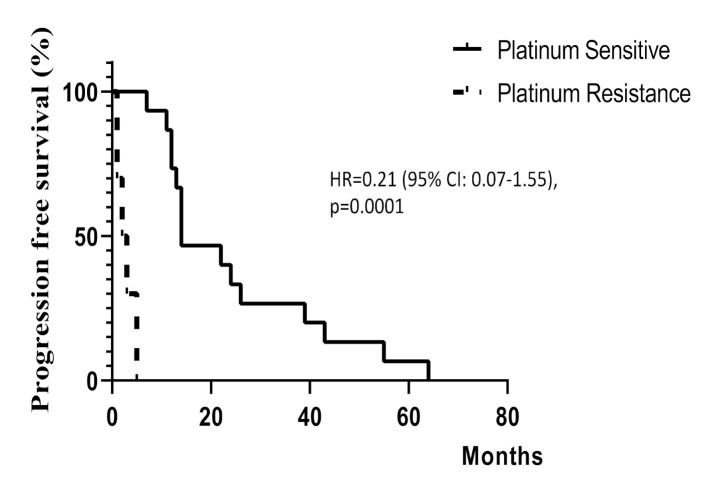
Kaplan–Meier curves of progression-free survival interval between the PS (*n* = 14) and platinum-resistant (*n* = 11) HGSC patients.

**Figure 4 diagnostics-11-00526-f004:**
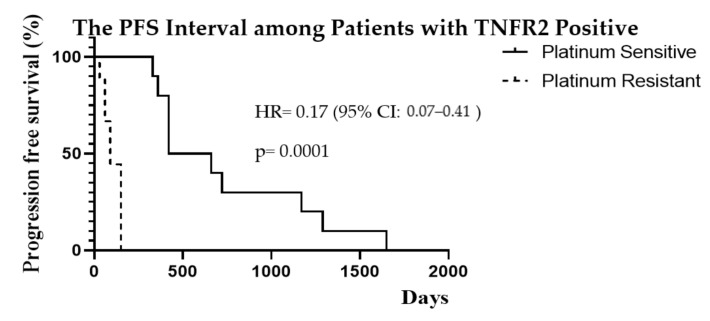
Kaplan–Meier curves of PFS interval in the PS (*n* = 14) and platinum-resistant group (*n* = 11) among patients with positive TNFR2 expression.

**Figure 5 diagnostics-11-00526-f005:**
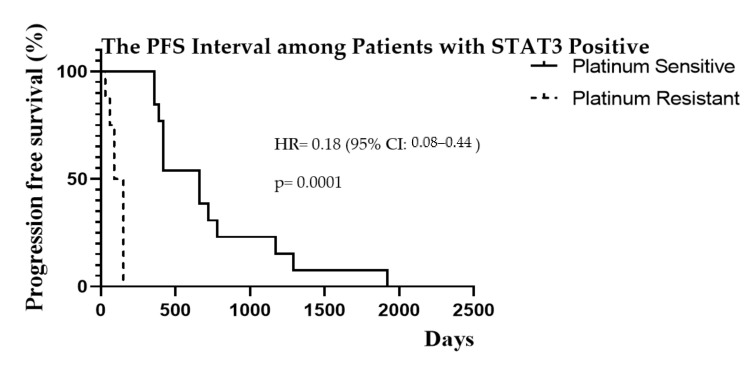
Kaplan–Meier curves of PFS interval in the PS (*n* = 14) and platinum-resistant group (*n* = 11) among patients with positive STAT3 expression.

**Table 1 diagnostics-11-00526-t001:** Primary Antibodies used in Immunohistochemistry.

No.	Primary Antibody	Clonality [Clone Number]	Product Code/Source	Primary Antibody Dilution	Incubation Duration	Positive Tissue Control
1.	Anti-TNF Receptor II antibody	Rabbit monoclonal [EPR1653]	ab109322/abcam Cambridge UK	1:100	60 min	Kidney tissue
2.	Anti-STAT3 antibody	Rabbit monoclonal [EPR787Y]	ab68153/abcam Cambridge UK	1:200	30 min	Pancreas tissue

**Table 2 diagnostics-11-00526-t002:** Demographic data of study population.

Demographics	Platinum Sensitive	Platinum Resistant	*p*-Value
^a^ Age at diagnosis in years, median (IQR)	51.0 (21.0)	59.0 (6.0)	0.008
^a^ Age of menarche in years, median (IQR)	13.0 (2.0)	13.0 (2.0)	0.668
^a^ Parity, median (IQR)	3.0 (2.0)	3.0 (1.0)	0.134
^c^ Body Mass Index, *n* (%)			
Underweight	1.0 (7.1)	1.0 (9.1)	
Normal BMI	5.0 (35.7)	2.0 (18.2)	
Overweight	6.0 (42.9)	7.0 (63.6)	
Obese	2.0 (14.3)	1.0 (9.1)	
^b^ Menopausal state, *n* (%)			
Yes	7.0 (50.0)	11 (100.0)	0.008
No	7.0 (50.0)	0 (0)	
^b^ Family History of ovarian, breast or colon cancer, *n* (%)			
Yes			
No			
^b^ Personal History of breast or colon cancer, *n* (%)			0.440
Yes	0 (0)	1.0 (9.1)	
No	14.0 (100)	10.0 (90.9)	
^d^ FIGO stage, *n* (%)			
I	4.0 (28.6)	1.0 (9.1)	0.341
II	2.0 (14.3)	0(0)	
III	7.0 (50.0)	5.0 (45.0)	
IV	1.0 (7.1)	5.0 (45.0)	
^a^ Pre-operative serum CA125, median (IQR)	848.0 (2049.0)	577.0 (3350)	0.811
Debulking surgery, *n* (%)			
Optimal	12 (87.5%)	8 (72.7%)	
Suboptimal	2 (14.3%)	3 (27.3%)	
^a^ Progression free survival in days (PFS), median (IQR)	540.0 (817.5)	90.0 (120.0)	

^a^ Independent *t*-test, ^b^ Fisher’s Exact Test, ^c^ Chi-squared test, ^d^ Logistic regression. IQR: interquartile range.

**Table 3 diagnostics-11-00526-t003:** TNFR2 protein expression of study population.

^b^ TNF2 Protein Expression, *n* (%)	Platinum Sensitive	Platinum Resistant	*p*-Value
Negative	0 (0)	0 (0)	0.67
Weak Positive	4.0 (28.6)	2.0 (18.2)	
Strong Positive	10.0 (71.4)	9.0 (81.8)	

^b^ Fisher’s Exact Test.

**Table 4 diagnostics-11-00526-t004:** STAT3 protein expression of study population.

^b^ STAT3 Protein Expression, *n* (%)	Platinum Sensitive	Platinum Resistant	*p*-Value
Negative	0 (0)	0 (0)	1.00
Weak	3.0 (21.4)	3.0 (27.3)	
Strong	11.0 (78.6)	8.0 (72.7)	

^b^ Fisher’s Exact Test.

**Table 5 diagnostics-11-00526-t005:** TNFR2 and STAT3 expression at different stages of ovarian cancer.

^d^ FIGO Staging	TNFR2 Expression			STAT3 Expression		
	Weak	Strong	*p*-Value	Weak	Strong	*p*-Value
IC	0 (0)	5.0 (26.3)	0.32	3.0 (50.0)	2.0 (10.5)	0.53
II	0 (0)	2.0 (10.5)		0 (0)	2.0 (10.5)	
III	3.0 (50.0)	9.0 (47.4)		2.0 (33.3)	10.0 (52.6)	
IV	3.0 (50.0)	3.0 (15.8)		1.0 (16.7)	5.0 (26.4)	

^d^ Logistic regression.

**Table 6 diagnostics-11-00526-t006:** Association of TNFR2 and STAT3 expression with PFS.

Progression Free Survival (PFS)	^e^ TNFR2 Expression		*p*-Value	^e^ STAT3 Expression		*p*-Value
	Weak	Strong		Weak	Strong	
PFS in months median (IQR)	8.50 (31)	7.0 (8.0)	0.74 HR 1.40 (95% CI 0.51–3.84)	7.0 (34)	8.0 (18.0)	0.69 HR 0.92 (95% CI 0.30–2.80)

^e^ Kaplan–Meier survival curve logrank analysis, HR = hazard ratio.

## Data Availability

We declared that materials described in the manuscript, including all relevant raw data, will be freely available to any scientist wishing to use them for non-commercial purposes, without breaching participant confidentiality.

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
