# Peer review of "Prognostic Value of TNFR2 and STAT3 among High-Grade Serous Ovarian Cancer Survivors According to Platinum Sensitivity"

_diagnostics, 2021, doi:10.3390/diagnostics11030526_

Round 1

Reviewer 1 Report

Some sentences are plagairized. For example, "Given the heterogeneity of this disease, increases in long term survival might be achieved by translating recent insights at the molecular and cellular levels to personalize individual strategies for treatment and to optimize early detection. "May | 2009 | Ovarian Cancer. https://ovariancancerinfo.wordpress.com/2009/05/

Also too much reliance on online sources, such as Wikipedia. Need to use the primary sources of information from government agencies, like the World Health Organization, Centers for Disease Control, etc.

The first references in the manuscript, e.g. #1-3, are not ideal. These come from recently published scientific articles, but they should be primary sources of the statistics and information. In addition, there is too much reliance on #3.

It's very difficult to differentiate between the platinum sensitive and resistant lines in the graph (figure 3). Please enhance the dashed lines or color one line. Figure 8 looks fine - mimic this one for figure 3.

The platinum resistant group has more women at later stages, III and IV, which likely accounts for some of these differences. The authors should note this.

Too much over-interpretation to immunosuppression and the immune system. The study looks at staining in relation to platinum resistance, not immunotherapy. Minimize the extrapolation with TNFR2 and immunosuppression.

May | 2009 | Ovarian Cancer. https://ovariancancerinfo.wordpress.com/2009/05/May | 2009 | Ovarian Cancer. https://ovariancancerinfo.wordpress.com/2009/05/

Author Response

Reviewer 1 (Round 1)

No.

Comments from Editor

Response from the Authors

Location of the change in the manuscript

1.

Some sentences are plagiarized. For example, "Given the heterogeneity of this disease, increases in long term survival might be achieved by translating recent insights at the molecular and cellular levels to personalize individual strategies for treatment and to optimize early detection. "

The manuscript have gone through the Turnitin check for plagiarism and we have removed any plagiarised sentences and kept to acceptable percentage. 

Entire manuscript.

2.

Much reliance on online sources, such as Wikipedia. Need to use the primary sources of information from government agencies, like the World Health Organization, Centres for Disease Control, etc.

We have looked through the references used and have replaced all online sources with the a primary sources information.  

All references have been updated to cite primary sources as far as possible.

3.

The first references in the manuscript, e.g. #1-3, are not ideal. These come from recently published scientific articles, but they should be primary sources of the statistics and information. In addition, there is too much reliance on #3.

We have replaced references 1-3 with the original statistical study and information.

There is only one cited information from reference number 3.

1.Introduction

-Line 35-37.

4.

It's very difficult to differentiate between the platinum sensitive and resistant lines in the graph (figure 3). Please enhance the dashed lines or color one line. Figure 8 looks fine - mimic this one for figure 3.

Figure 3 has been re-enhanced with dashed lines and thicker line.

Line 291-297

5.

The platinum resistant group has more women at later stages, III and IV, which likely accounts for some of these differences. The authors should note this.

This information has been clearly highlighted and interpreted in the manuscript.

3.Results

Line 380-382

6.

Too much over-interpretation to immunosuppression and the immune system. The study looks at staining in relation to platinum resistance, not immunotherapy. Minimize the extrapolation with TNFR2 and immunosuppression.

Thank you for highlighting this point.

This section has been removed from the manuscript.

4.Discussion  Line 382-404

Reviewer 2 Report

Comments to the Authors
Thank you very much for providing me the chance to review this article. In general is an interesting article but there are some major issues that I would like the authors to clarify/revise.

  1. The authors do not report any inclusion or exclusion criteria in the M and M section. I think that even if this is a retrospective study the selection of the population have to be reported considering also that they enrolled only 25 patients in 8 years.
  2. The authors reported that all pts received adjuvant platinum based chemotherapy. On the other hand it is very important to know the chemotherapy dosage, any maintenance therapy have been proposed ? …. Considering that the last patients have been enrolled in 2019 , none received any additional target therapy ? every patient complete all 6 cycles ? no adverse effect have been reported.
  3. How recurrence have been evaluated ? imaging ? surgery ? Did the author use RECIST criteria?
  4. No standardized follow up method has been reported. I Think that if their end point is PFS the follow up strategy have to be clearly reported.
  5. I think that the last paragraph ( line 117) of the M and M section that reported how many patients was enrolled and the reasons of exclusion has to be reported in the result section
  6. Only 16 pts out of 25 received optimal surgery. I think that this rate is to low , considering that 5 pts out 25 were I figo stage. This is an important bias in the study and have to be clearly reported.
  7. I think that including 5 early stage OC ( 20%) the data on PFS may be influenced by the low rate of recurrence of this subgroup. Please comment on that.
  8. I think that BRCA status has to be reported and should be evaluated in relation to PFS and other markers.
  9. The number of the patients is to low considering the enrolment period (8 Yeas). Please comment on that
  10. Considering that the authors analysed only 25 patients, I think that the additional subclassification in platinum sensitive and resistant group led to a very low sample size. The PFS curves among patients with TNFR2 positive compare 4 vs 10 vs 9 vs 2 pts ( figure9). The same for PFS curves in STAT3 Positive compare 11 vs 3 vs 8vs 3 (figure 10) . I think that considering the small sample size the article could be focused only in the general population comaping PFS according to the markers results

Author Response

Reviewer 2 (Round 1)

No.

Comments from Editor

Response from the Authors

Location of the change in the manuscript

1.

The authors do not report any inclusion or exclusion criteria in the M and M section. I think that even if this is a retrospective study the selection of the population have to be reported considering also that they enrolled only 25 patients in 8 years.

The inclusion and exclusion criteria of the study subjects have been added.

The exclusion criteria of this study were very strict to ensure that no immune markers or other medical factors may affect the outcome of this research study.

2. Materials and Methods

Line 116-120.

2.

The authors reported that all pts received adjuvant platinum based chemotherapy. On the other hand it is very important to know the chemotherapy dosage, any maintenance therapy have been proposed ? …. Considering that the last patients have been enrolled in 2019 , none received any additional target therapy ? every patient complete all 6 cycles ? no adverse effect have been reported.

Detailed information on the chemotherapy treatment, dosage and maintenance therapy have been added.

2. Materials and Methods

Line 106-118.

3.

How recurrence have been evaluated ? imaging ? surgery ? Did the author use RECIST criteria?

Information on the recurrence evaluation and progressive disease (PD) have been highlighted.  

2. Materials and Methods

Line 106-129.

4.

No standardized follow up method has been reported. I Think that if their end point is PFS the follow up strategy have to be clearly reported.

Thank you for highlighting this point.

The follow up strategy performed for all the patients is clearly reported in the manuscript. 

2. Materials and Methods

Line 120-129.

5.

I think that the last paragraph ( line 117) of the M and M section that reported how many patients was enrolled and the reasons of exclusion has to be reported in the result section

Detailed explanation on the initial recruitment number of the study patients and the excluded numbers with the final total number of the investigated patients was reported in the result section.

3. Results

Line 228-231

6.

Only 16 pts out of 25 received optimal surgery. I think that this rate is to low , considering that 5 pts out 25 were I figo stage. This is an important bias in the study and have to be clearly reported.

Thank you for the highlighted. There is error in the statistics reported. We have corrected the error.

3. Result

Table 1. Line 281-283

4.Discussion

Line 376-378

Reviewer 3 Report

I read your paper with great interest as the need for biomarkers in HGSC is indeed high. I have some concrete recommendations to improve the quality of your work:

  • Your findings are clear and well described but I would recommend that you enlist a professional proofreader to edit your article.
  • I would suggest to drop the 'study procedure flow chart' as it doesn't add to the comprehensibility but rather gives the impression of a student paper
  • Maybe you could try to eloborate more about the unexpected correlation between TNFR2/STAT3 strong expression group and progression free survival.

Author Response

Reviewer 3 (Round 1)

No.

Comments from Editor

Response from the Authors

Location of the change in the manuscript

1.

·       Your findings are clear and well described but I would recommend that you enlist a professional proofreader to edit your article.

We have assigned the university proofreading service to correct the manuscript.

Entire manuscript

2.

·       I would suggest to drop the 'study procedure flow chart' as it doesn't add to the comprehensibility but rather gives the impression of a student paper

Thank you for the notification. We agreed to remove the procedure flow chart from manuscript.

Flow Chart has been removed. Information has been updated in methodology

3.

·       Maybe you could try to eloborate more about the unexpected correlation between TNFR2/STAT3 strong expression group and progression free survival.

We have decided to remove this sub-analysis from the manuscript as it may be biased due to the small sample size, as suggested by another reviewer. We hope in the future to do this analysis with large sample size, so we can validate this findings.

Remove from result and discussion

Round 2

Reviewer 2 Report

I think that the manuscript could be accepted